# The Role of Oxidative Stress in Kidney Injury

**DOI:** 10.3390/antiox12091772

**Published:** 2023-09-16

**Authors:** Nejc Piko, Sebastjan Bevc, Radovan Hojs, Robert Ekart

**Affiliations:** 1Department of Dialysis, Clinic for Internal Medicine, University Medical Centre, 2000 Maribor, Slovenia; robert.ekart2@guest.arnes.si; 2Department of Nephrology, Clinic for Internal Medicine, University Medical Centre, 2000 Maribor, Slovenia; sebastjan.bevc@ukc-mb.si (S.B.); radovan.hojs@guest.arnes.si (R.H.); 3Medical Faculty, University of Maribor, 2000 Maribor, Slovenia

**Keywords:** acute kidney injury, acute tubular necrosis, chronic kidney disease, oxidative stress, reactive oxygen species

## Abstract

Acute kidney injury and chronic kidney disease are among the most common non-communicable diseases in the developed world, with increasing prevalence. Patients with acute kidney injury are at an increased risk of developing chronic kidney disease. One of kidney injury’s most common clinical sequelae is increased cardiovascular morbidity and mortality. In recent years, new insights into the pathophysiology of renal damage have been made. Oxidative stress is the imbalance favoring the increased generation of ROS and/or reduced body’s innate antioxidant defense mechanisms and is of pivotal importance, not only in the development and progression of kidney disease but also in understanding the enhanced cardiovascular risk in these patients. This article summarizes and emphasizes the role of oxidative stress in acute kidney injury, various forms of chronic kidney disease, and also in patients on renal replacement therapy (hemodialysis, peritoneal dialysis, and after kidney transplant). Additionally, the role of oxidative stress in the development of drug-related nephrotoxicity and also in the development after exposure to various environmental and occupational pollutants is presented.

## 1. Introduction

Kidney disease is an important worldwide health problem, leading to increased morbidity and mortality. Additionally, it is also associated with high economic burden. For patients with chronic kidney disease (CKD), the leading cause of mortality is cardiovascular disease [1]. Traditional cardiovascular risk factors, often present in CKD patients, are diabetes, hypertension, and dyslipidemia. However, several additional risk factors are specific for patients with kidney disease and cause an exponential increase in cardiovascular risk. One of the major kidney disease-specific risk factors is an imbalance between the overproduction of reactive oxygen species (ROS) and/or a reduction in antioxidant defense mechanisms. Oxidative stress is potentially implicated in endothelial injury, atherosclerosis, atherosclerotic plaque rupture, cardiovascular events, and also in CKD progression [2]. This review article, therefore, illuminates the role of oxidative stress throughout different spectrums of kidney disease and also in the development of kidney injury after the exposure to various environmental and occupational pollutants. 

## 2. Literature Search

The Preferred Reporting Items for Systematic Reviews and Meta-Analyses (PRISMA) 2020 [3] was used in preparation of the manuscript. When searching for references, we used multiple databases (PubMed, PubMed Central, Cochrane, and MEDLINE). The keywords kidney disease, acute kidney injury, chronic kidney disease, and oxidative stress were used when searching for appropriate literature. 

## 3. Acute Kidney Injury

Acute kidney injury (AKI) is a syndrome defined by a rapid increase in serum creatinine, reduced urine output (oliguria or anuria), or both. It is an essential complication in hospitalized patients, occurring in 10–15% of all hospitalizations [4]. It is especially prevalent in intensive care units, where the prevalence can reach 50% or more [5]. Studies have corroborated prolonged hospital stay, increased likelihood of developing CKD, and increased short- and long-term mortality in hospitalized patients with AKI [6]. Furthermore, the global burden of AKI-related mortality exceeds that of breast cancer, heart failure, or diabetes mellitus, with mortality remaining high despite numerous efforts and advancements in diagnosing and managing patients with this condition [7]. Several environmental, socioeconomic, and patient-related risk factors for the development of AKI are summarized in Figure 1 [8].

It should be remembered that AKI is a clinical term and can be due to glomerular or tubulointerstitial involvement or both. In AKI, tubular involvement is more common and is the focus of this article. Pathologists use descriptive pathological findings, often called acute tubular injury (ATI). Prerenal, intrarenal, and postrenal causes of AKI can all cause different degrees of ATI. Dissociation between clinical and pathological findings is not rare, especially in prerenal causes of AKI, for example, hypovolemia and hemorrhagic shock. In such cases, ATI is mild or even absent [9]. At the same time, patients are often dialysis-dependent and experience several complications related to kidney failure, for example, fluid overload, electrolyte disturbances, metabolic acidosis, and uremia [10]. 

ATI is characterized by focal or diffuse tubular luminal dilatation, simplification of the lining epithelium, loss of the brush border in proximal tubules, loss of nuclei, and the presence of nucleoli. At times, signs of epithelial cell regeneration are present (epithelial cell mitosis and cytoplasmic basophilia). In severe cases, especially in prolonged AKI, focal or diffuse tubular cell necrosis is present. In these instances, the term acute tubular necrosis (ATN) is used [11]. 

Oxidative stress is crucial in developing kidney injury, ranging from AKI to irreversible CKD [12] (Figure 2). In this review article, some data on the role of oxidative stress in the development of kidney injury is presented. 

## 4. Oxidative Stress

### 4.1. The Definition and Significance of Oxidative Stress

Oxidative stress is the imbalance favoring the increased generation of ROS and/or reduced body’s innate antioxidant defense mechanisms [13]. The processes of oxidation and reduction constantly happen in the cells, but the damage does not usually occur due to several protective enzymatic and non-enzymatic anti-oxidative mechanisms that keep both of these processes in balance. Whenever there is a balance shift towards increased oxidation, oxidative stress occurs [14]. A pathological shift towards oxidative stress and consequent cell and tissue injury results in the modification of lipid, protein, and DNA structures, impacting their functions as well [15]. In addition to causing direct tissue damage, oxidative stress can activate multiple intracellular signaling pathways and thereby indirectly induce cell apoptosis or cell overgrowth, extracellular matrix production and degradation, oxygen sensing, and inflammation, leading to significant organ dysfunction of the cardiovascular system, pancreas, kidneys, and lungs [16,17]. 

Direct analysis of free radicals in an in vivo system is demanding due to their short lifespan. Additionally, measuring antioxidants is challenging, primarily because many antioxidants with different properties exist. A significant limitation is that oxidative stress is usually a focal response to tissue injury, making detecting it even more taxing. Due to this, these measurements are complex, expensive, and time-consuming [17]. Studies have thus far used a plethora of different techniques of antioxidative capacity and oxidative stress, for example, total antioxidant capacity (TAC) and ferric-reducing ability of plasma (FRAP), epoxidation end-products measurements [18], carbonyl derivates of amino acid residues (lysine, proline, threonine, and arginine), advanced oxidation protein products (AOPP) [19], advanced glycation end products (AGEs) [20], 8-oxo-2′-deoxyguanosine as a marker of oxidative DNA damage [21], and 7,8-dihydro-8-deoxyguanosine as a marker of oxidative RNA damage [22]. It is pivotal to point out that different features should be used to fully assess the extent of oxidative damage in the body instead of relying on a single marker [18].

### 4.2. Oxidative Stress and Chronic Kidney Disease

Oxidative stress is commonly found in CKD and is associated with increased all-cause mortality in these patients. It is already present in the early stages of CKD and is even more apparent in patients on hemodialysis (HD) and peritoneal dialysis (PD) [23,24]. Moreover, oxidative stress has also been observed in patients after renal transplantation [25]. 

During the last two decades, oxidative stress has been at the center of attention as a novel, non-traditional risk factor for inflammation, atherosclerosis, diabetes mellitus (DM), and CKD progression. The heavy cardiovascular burden in patients with CKD is at least partly due to the oxidative stress observed in these patients [26]. 

ROS in the kidneys is mainly produced by the mitochondrial respiratory chain and enzymes, such as Nicotinamide Adenine Dinucleotide Phosphate (NADPH) oxidase (NOX). The different NOX isoforms, including NOX1, NOX2, and NOX4, are associated with oxidative stress, worsening vascular function, and promoting interstitial fibrosis [27]. Recently, NOX5 expression was found to be increased in human biopsy samples of patients with diabetic nephropathy as well [28]. Furthermore, an increase in NOX5-derived ROS is essential for the faster progression of diabetic nephropathy [29]. 

Additional oxidative markers in CKD are malondialdehyde (MDA), oxidized low-density lipoprotein (LDL), AGEs, and 8-hydroxyde-oxyguanosine [30]. A decisive pathogenetic mechanism of severe cardiovascular burden in CKD patients appears to be the increased production of angiotensin II, which can cause an increase in the expression of NOX1 and NOX2 and a decrease in anti-oxidative defense mechanisms. Furthermore, a subset of antioxidant enzymes, the paraoxonases (PON), deserve special attention due to abundant clinical evidence regarding reduced serum PON1 activity in CKD as a contributor to the increased burden of cardiovascular disease [31]. The increased production of ROS, especially O_2_^-^ leads to decreased production of nitric oxide (NO), an essential antioxidant protecting kidney function by increasing renal blood flow, enhancing pressure natriuresis, regulating tubuloglomerular function, and preserving fluid and electrolyte homeostasis [32]. Studies have also shown a downregulation in the expression of catalase, superoxide dismutase (SOD), and NADPH dehydrogenase quinone 1 in the CKD population, all of which are important anti-oxidative enzymes [25]. 

Diabetic kidney disease (DKD) is one of the most common causes of end-stage renal disease and is also associated with severely increased morbidity and mortality. The pathogenesis and clinical manifestations of DKD usually follow a predetermined course, beginning with albuminuria, advancing to overt proteinuria, and ultimately leading to a progressive reduction in the estimated glomerular filtration rate (eGFR). Accumulating evidence has demonstrated the overproduction of ROS as the common denominator linking the altered metabolic pathways in the kidney with disrupted renal hemodynamics known to be associated with DKD [29]. Renal ROS production in DKD is predominantly mediated by various NOXs and a consequent increase in O_2_^-^, H_2_O_2_, and OH. A defective antioxidant system is also important. It appears that ROS production is present not only in the podocytes and the glomerulus but also in the tubulointerstitium and is associated with pronounced and enhanced inflammation and fibrosis [33].

Autosomal dominant polycystic kidney disease (ADPKD) is the most common hereditary kidney disease. Renal manifestations of ADPKD are gradual cyst development and kidney enlargement, ultimately progressing to end-stage renal disease. Patients with ADPKD have increased serum levels of asymmetric dimethyl arginine (ADMA), MDA, and oxidized LDL cholesterol, compared to the healthy cohort. Additionally, their antioxidant capacity is reduced, which is evident from reduced plasma SOD concentrations. It appears oxidative stress plays an important role in the development and progression of kidney disease in these patients, as well [34]. 

A major characteristic of focal segmental glomerulosclerosis (FSGS) is podocyte injury. Studies have shown podocyte injury can be at least partly attributed to oxidative stress. It appears that an increase in transforming growth factor beta (TGF-β) induces endothelin synthesis and oxidative stress in glomerular endothelium. Mitochondrial oxidative DNA damage was found even before podocyte injury [12,35].

In patients with HD, increased concentrations of oxidative stress biomarkers have been found. Some of them are due to comorbidities and conditions accompanying these patients, such as dyslipidemia, arterial hypertension, old age, metabolic syndrome, DM, and atherosclerosis [36]. Secondly, antioxidant mechanisms in these patients are impaired, as shown by reduced TAC and oxidized glutathione and reduced levels of vitamins E, C, and D [37]. The use of vitamin E-coated dialyzer can decrease ROS in HD patients, reflected by the decrease in serum C-reactive protein (CRP) and interleukin-6 (IL-6) [38]. Thirdly, the chronic inflammation found in HD patients is directly related to oxidative stress [38].

Moreover, the HD procedure is interlinked with activating prooxidative mechanisms (for example, increased intracellular ROS, NO, markers of protein, lipid, and DNA oxidation) and the loss of antioxidative mechanisms [39]. HD-related factors that cause the excessive generation of ROS are bioincompatible dialyzer membranes, heparin-based anticoagulation, prolonged use of central venous catheters instead of arteriovenous fistulas, standard HD instead of hemodiafiltration, longer duration of HD sessions, contaminated dialysate, and certain administered medications, namely, intravenous iron and erythropoietin stimulating agents [37]. Studies have shown 14 times higher values of ROS after an HD session compared to the values before HD, partly due to ROS generation and partly due to antioxidative insufficiency [40]. It appears that ROS generation begins shortly after starting HD (after 15 min) and then returns to its predialysis level by the end of the HD session [41]. 

PD is more biocompatible than HD, resulting in less oxidative stress. However, PD patients still manifest oxidative stress compared to the general population and predialysis CKD patients, mainly due to the composition of the PD solution (high-glucose content, low pH, elevated osmolality, increased lactate concentration, and glucose degradation products). In addition to apparent detrimental cardiovascular effects, oxidative stress in PD patients is linked to the loss of residual renal function, peritoneal fibrosis, and ultimately, the development of encapsulating peritoneal sclerosis [24,42]. 

Kidney transplantation leads to lower levels of ROS and a better preservation of anti-oxidant capacity compared to HD or PD, which is most likely one of the reasons why these patients have improved survival compared to patients with HD or PD [43]. However, in kidney transplantation, factors such as immune response to allograft, ischemia/reperfusion injury, infections, and immunosuppressive therapy can be a source of significant oxidative stress [44]. They can cause graft tissue damage due to the loss of nephrons and the formation of fibrosis [45]. Measuring oxidative stress markers, such as MDA, is promising in predicting allograft survival and delayed graft function [46]. Additional markers of oxidative stress in this population include low molecular weight advanced glycation end products (LMW-AGEs), lipid peroxidation products, advanced oxidation protein products, and plasma ADMA [46].

## 5. The Role of Oxidative Stress in Acute Kidney Injury

Conventionally, acute kidney injury (AKI) has been viewed in an anatomical context (prerenal, renal, and postrenal), whereas, at the same time, it has been considered a risk factor for the development of subsequent CKD [47]. Several pathophysiologic advances have been made in recent years that help us better understand the reciprocal interconnection between AKI and CKD. Many of these factors involve oxidative stress [47,48]. 

The most important causes of AKI, especially ATN, are linked to ischemia and hypoxia. The kidneys have a sensitive system of blood flow autoregulation, which causes the GFR to be stable throughout different blood pressures. Two main mechanisms involved in maintaining renal autoregulation are tubuloglomerular feedback and myogenic response [14]. 

In addition to nitrogen waste products, AKI is associated with elevated levels of indole and carbonyl compounds, which can upregulate systemic oxidative stress [39]. NO is a vasodilator molecule formed by NO synthase (NOS), one of the most critical molecules in renal autoregulation. NO causes the vasodilatation of the afferent arteriole, leading to an increase in GFR, and it is also involved in several antifibrotic and anti-apoptotic pathways in the kidney [49]. Intracellularly, low levels of NO inhibit cytochrome c-oxidase, altering the generation of ROS in the mitochondria, upregulating the hypoxia-inducible factor (HIF) in endothelial cells, and stimulating the nuclear factor erythroid 2-related factor 2 (Nrf-2). HIF and Nrf2 are renoprotective and protect the renal tissues against oxidative stress [50,51]. Higher levels of NO interact with bound iron and can produce NO-derived reactive nitrogen species that can nitrosate thiols and lead to cell damage [51]. 

Hall et al. studied mitochondrial structure, function, and oxidative stress in rat kidneys in response to ischemia [52]. Their results showed changes in mitochondrial Nicotinamide Adenine Dinucletoide (NADH) and proton levels, followed by an upregulation in mitochondrial O_2_ and disjointed mitochondria, concluding that mitochondrial dysfunction is an essential step in the early phase of ischemic renal injury [52]. During ischemia, the spike in ROS drives the peroxidation of cardiolipin, a change thought to distort cristae and thereby impair efficient oxidative phosphorylation [53]. In a study by Sureshbabu et al., the authors found that increased levels of ROS in AKI induce enhanced selective mitochondrial apoptosis, leading to cell injury [54]. Mitochondrial dysfunction causes an increase in ROS, which, in turn, causes microvascular dysfunction as well, enhancing the effect of hypoxia on the renal medulla [55]. An additional study by Tanaka et al. found changes at the lysosomal level in the renal epithelium with the loss of brush border in the proximal tubule after exposure to gentamycin, followed by an increase in markers of oxidative stress and alterations in NADH levels [56]. 

The superoxide anion (O_2_^-^) is generated by oxidases or inside mitochondria through the electron transport chain. Local ischemia and cytokines induced by AKI (especially in AKI associated with sepsis) activate the endothelium of renal vasculature and recruit cells from the immune system that can generate O_2_^-^ via NOX [57,58]. 

Additional ROS found to be involved in AKI are hydrogen peroxide (H_2_O_2_) generated by dismutation or by oxidases from molecular oxygen, causing diverse cell injury and also reacting with iron-containing molecules, releasing more ROS [14]; hydroxyl radical (OH) generated by the Fenton reaction, causing lipid peroxidation and an increase in other ROS [42]; and hypochlorous acid (HClO), which is generated by local inflammatory cells and causes a reaction with amines, leading to the formation of toxic chloramines [14,57]. Lipid peroxidation is also enhanced by the heme group in myoglobin in the case of rhabdomyolysis-induced AKI [59,60].

In a study by Seija et al., the mechanisms of nitrosative stress were tested on rat kidneys in sepsis-induced AKI [61]. The authors found increased protein nitration and upregulation in NOS-1 and NOS-2. They concluded that peroxynitrites are vital in developing sepsis-induced AKI [61]. In Gram-negative sepsis, lipopolysaccharide (LPS) selectively binds to the Toll-like receptor (TLR), which is present in several cells, including renal tubules. The binding of LPS to TLR4 triggers a cascade reaction of events that leads to enhanced oxidative stress. Tumor necrosis factor α (TNF-α) and interleukin-1β (IL-1β) promote the release of H_2_O_2_, with oxidative damage that further potentiates the inflammatory response [62]. Several studies have shown the upregulation of TLR4 in sepsis, increasing the potential burden of selective LPS binding, consequent inflammation, and oxidative stress [63,64]. Activating other receptors (for example, TLR9 receptors, with the binding of mitochondrial and bacterial DNA) can also lead to ROS production and oxidative stress [65].

Oxidative stress is one of the most important causes of delayed graft function in patients after kidney transplant, most commonly due to ischemia–reperfusion injury. Immediately after reperfusion, a sudden increase in O_2_^-^ production inside mitochondria is noticed, leading to neutrophil activation and inflammation [66]. In a study by Tanaka et al., the authors found expression of adhesion protein-1 in pericytes, a critical protein that can generate H_2_O_2_ and activate inflammatory cells [56]. A decrease in anti-oxidant capacity has been found in mice with the loss of Nrf2 and heme oxygenase-1 [67]. Additionally, mice deficient in transient receptor potential melastatin-2 (TRPM-2) showed resistance to oxidative stress and apoptosis. The authors found an increase in TRPM-2 expression after renal ischemia, further indicating that oxidative stress and renal ischemia are closely linked [68].

## 6. Biomarkers of Oxidative Stress in AKI

Ware et al. studied 50 critically ill patients with severe sepsis. They found that plasma F2-isoprostane and isoflurans were higher in patients who later developed AKI and acute hepatic failure. They postulated that lipid peroxidation is crucial to sepsis-induced multiorgan failure [69]. 

In a prospective observational study by Costa et al., 132 critically ill patients were included [70]. Their results showed a potential role of decreased erythrocyte SOD-1 in the development of AKI [70].

NGAL (neutrophil gelatinase-associated lipocalin) can be measured in serum and urine and is a marker of acute tubular injury. It has bacteriostatic and anti-oxidative properties. An induction of urine NGAL under harmful conditions (such as oxidative stress) is a compensatory response to ameliorate oxidative stress-mediated toxicity. It is, therefore, a good indirect predictor of oxidative stress and also a marker of AKI, especially when compared to serum creatinine, which has several drawbacks in detecting AKI [71]. 

According to Tanase et al., urinary kidney injury molecule-1 (KIM-1) indicates acute kidney injury due to tubulointerstitial injury and can also be used in certain cases of xenobiotic nephrotoxicity, such as cisplatin-induced nephrotoxicity. It is important to emphasize that it is a non-specific marker of tubular injury and can be elevated due to a plethora of causes, including oxidative stress [72]. A meta-analysis by Kokkoris et al. showed the importance of additional markers of AKI and oxidative stress, for example, L-FABP (Liver-Type Fatty Acid Protein) and N-acetyl-ß-D-glucosaminidase (NAG) [71]. Urinary insulin-like growth factor-binding protein 7 (IGFBP7) and tissue inhibitor of metalloproteinases-2 (TIMP-2) are increasingly recognized as the best markers of developing AKI in septic patients. These markers are cell cycle arrest biomarkers as they have been implicated in the G1 cell cycle phase during the early stage of oxidative cell stress [73].

Thioredoxin (Trx) protects proteins and cell membranes from oxidative injury. In addition to its antioxidant effects, it also regulates cytokine transcription and apoptosis. In ROS-induced cardiomyocyte injury, Trx clears ROS and enhances mitochondrial function, preventing cardiomyocyte hypertrophy and apoptosis. There are also some data on the antioxidant and protective effects of Trx in preventing acute kidney injury [74]. 

In Table 1, the main biomarkers of oxidative stress in AKI are summarized.

## 7. Oxidative Stress, SGLT-2 Inhibitors and Non-Steroid Mineralocorticoid Receptor Antagonists

Sodium–glucose cotransporter-2 (SGLT-2) inhibitors are a new class of glucose-lowering agents. Extensive clinical trials have revealed several cardiac and renal protective effects of these drugs, even in non-diabetic patients, expanding their uses in clinical practice [75]. 

Several mechanisms underlying the renal benefits of SGLT-2 inhibitors have been proposed. They reduce high glucose-induced oxidative stress in the proximal tubule, decrease intraglomerular pressure via afferent arteriole modulation, decrease proteinuria and inflammation, ameliorate interstitial fibrosis, improve mitochondrial function, and lead to reduced sympathetic nervous system activation [76]. Cardioprotective effects include osmotic diuresis, natriuresis, improved endothelial function, lower serum uric acid levels, decreased markers of oxidative stress, and suppressed AGEs formation [77]. Several studies have, in detail, observed the anti-oxidative properties of SGLT-2 inhibitors. Van Bommel et al. observed reduced urinary excretion of 8-oxo-7,8-dihydro-2′-deoxyguanosine (8-oxo-dG), a DNA oxidation marker, in patients treated with dapagliflozin [78]. Lambadiari et al. performed a study on 160 type-2 DM patients with empagliflozin. They found that empagliflozin increased 2,2¢-azino-bis-(3-ethylbenzthiazoline-6-sulphonic acid) (ABTS)’s radical scavenging capacity, a measure of antioxidant ability and serum levels of the thiobarbituric acid reactive substances (TBARS) and MDA, indicators of lipid peroxidation [79]. The results of an observational pilot study by Nabrdalik-Lesniak et al. confirmed an improvement in SOD antioxidant defense in patients treated with an SGLT-2 inhibitor [80]. 

A new drug used for treating diabetic CKD patients—a non-steroid, selective mineralocorticoid antagonist, finerenone—also has several antioxidative properties. Preclinical studies on rats (treated with finerenone) have shown improved endothelial function due to enhanced NO bioavailability and decreased O_2_^-^ levels due to an upregulation in SOD activity. This finding was associated with increased renal SOD activity and reduced albuminuria [81]. Additionally, the expression of markers of tubular injury (such as KIM-1and NGAL) was decreased in rats treated with finerenone, and finerenone prevented an increase in MDA and 8-hydroxyguanosine after renal ischemia-reperfusion injury in human subjects [82].

## 8. Oxidative Stress and Drug-Induced Kidney Injury

The kidney is exposed to a higher concentration of drugs and their metabolites due to the secretion of ionic drugs through tubular organic ion transporters across the luminal membranes of renal epithelial tubular cells, and through the reabsorption of filtered toxic substances back into the lumen of the tubule. Drug-induced kidney injury is a serious and common cause of AKI in hospitalized patients, accounting for roughly 20% of cases. Usually, two common pathological entities are seen in these patients, acute interstitial nephritis (AIN) and ATN. While AIN is caused by allergic reactions, ATN is often caused by drug-induced oxidative stress [83]. 

An important pathway involved in these cases is the vanin-1 pathway. An inciting drug causes the generation of ROS. In the presence of oxidative stress, antioxidant response-like elements within the vinin-1 promoter region enhance vinin-1 expression. Consequently, more cysteamine is formed from the hydrolysis of pantheteine, and cysteamine is then converted to cystamine. Cystamine inhibits glutathione synthesis. Glutathione stores decrease, leading to an intensification of oxidative stress and enhanced production of inflammatory cytokines and chemokines [84]. It appears a defective antioxidant mechanism, with reduced glutathione stores, is crucial in the development of drug-induced AKI [85].

Moreover, several biomarkers of drug-induced oxidative stress have been found, including advanced oxidation protein products, protein reactive carbonyl derivatives, and methylglyoxal. A reduction in plasma catalase level have also been observed [86,87]. 

In Table 2, several drugs that have been associated with ATN due to oxidative stress are presented [83]. 

## 9. Occupational and Environmental Toxic Substances, Oxidative Stress, and Kidney Injury

Cadmium (Cd) is a nonessential metal that has heavily polluted the environment due to human activities. It has a tendency to accumulate within the proximal tubule and can induce dysfunction of the mitochondrial electron transport chain, leading to enhanced production of ROS and impaired function of NOX. Most important steps in Cd-induced nephrotoxicity are reabsorption and lysosomal degradation of Cd^2+^-metallothionein complexes in the proximal and also distal tubule, release of free Cd^2+^, perturbation of cellular calcium homeostasis, and interference with mitochondrial electron transport chain. Clinical consequences are partial or complete Fanconi syndrome and progressive kidney disease [88].

Additional constituents of electronic waste include lead (Pb), mercury (Hg), arsenic (As), and silica (SiO_2_). Similar to Cd, they also induce oxidative stress predominantly in the proximal tubule [89]. Interestingly, some of these nephrotoxic substances act in concert. For example, exposure to Pb enhances oxidative stress and nephrotoxicity of Cd, mainly through interference with mitochondrial function, cellular calcium homeostasis, and impaired function of NOX [89]. Exposure to Hg has been associated with lower serum glutathione peroxidase and with a higher titer of antibodies toward MPO; this was even more pronounced in smokers [90]. Patients with chronic exposure to As were found to have elevated markers of oxidative stress, for example, urinary MDA and 8-hydroxy-2′-deoxyguanosine [91]. SiO_2_ is weakly acidic and can cause renal injury through mesoporous SiO_2_ (MSNs) particles and oxidative stress [92]. 

Trichloroethylene (TCE) is an important organic solvent widely used as a metal part cleaner, dry cleaning agent, and chemical extractant. Experimental studies have demonstrated an increase in oxidative stress biomarkers in mice sensitized with TCE (an increase in MDA and a decrease in renal NO, NOS, and SOD concentrations) [93].

Hexachloro-1:3-butadiene (HCBD) is a halogenated solvent that is known to cause damage specific to pars recta of the renal proximal tubule in rats. Experimental studies have shown a proportional increase in urinary excretion of α-glutathione S-transferase (α-GST) and KIM-1, emphasizing the role of oxidative stress in HCBD-induced kidney injury [94].

## 10. Conclusions

One of the central factors for developing progressive kidney injury (either acute or chronic) is oxidative stress and ROS formation. It appears oxidative stress is also critical in increased cardiovascular risk observed in these patients. Additionally, nephrotoxicity of several medications and occupational and environmental pollutants are mediated through oxidative stress. Despite several new drugs in recent years (SGLT-2 inhibitors and non-steroid mineralocorticoid receptor antagonists), new treatment strategies explicitly targeting oxidative stress are required. Further studies are needed to conceptualize and expand our understanding of oxidative stress in developing kidney injury.

## Figures and Tables

**Figure 1 antioxidants-12-01772-f001:**
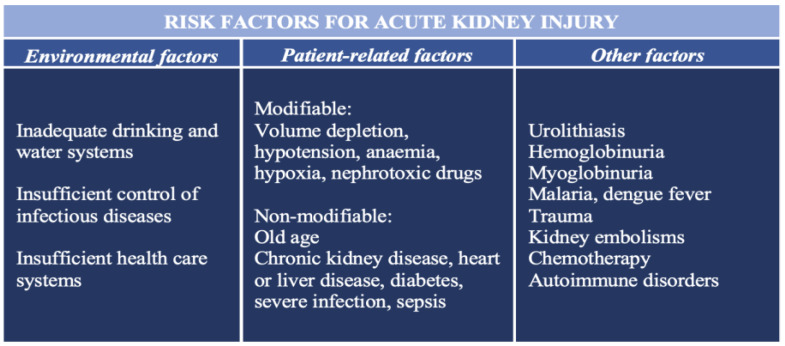
The most important risk factors for the development of acute kidney injury.

**Figure 2 antioxidants-12-01772-f002:**
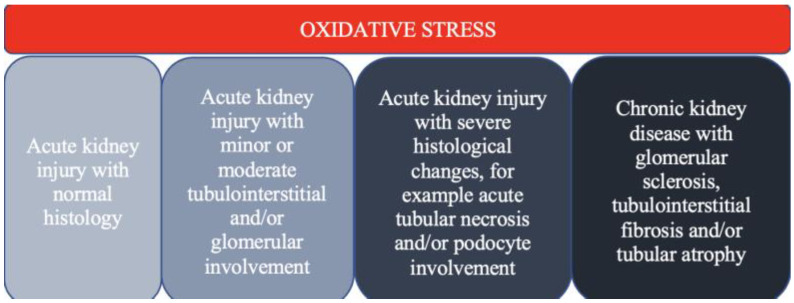
The spectrum of kidney injury in the context of oxidative stress.

**Table 1 antioxidants-12-01772-t001:** A summary of the main markers of oxidative stress in acute kidney injury (AKI).

Biomarker of Oxidative Stress in Acute Kidney Injury	The Role of the Biomarker	Author
Increased plasma F2-isoprostane, isoflurans	Markers of lipid peroxidation, associated with renal, hepatic, circulatory, and coagulation failure	Ware et al. [69]
Reduced Erythrocyte Superoxide Dismutase-1 (SOD-1)	Reduced SOD-1 is associated with increased reactive oxygen species	Costa et al. [70]
Increased urinary Neutrophil Gelatinase-Associated Lipocalin (NGAL) Increased Liver-Type Fatty Acid Protein (L-FABP) Increased urinary N-acetyl-ß-D-glucosaminidase (NAG)	Compensatory increase under oxidative stress L-FABP may be an important cellular antioxidant during oxidative stress, by maintaining low levels of free fatty acids in the cytoplasm of tubular cells. An increase in L-FABP is associated with oxidative stress Marker of tubulointerstitial injury due to different causes, including oxidative stress	Kokkoris et al. [71]
Increased urinary Kidney Injury Molecule-1 (KIM-1)	Marker of tubulointerstitial injury due to different causes, including oxidative stress	Tanase et al. [72]
Increased urinary Insulin-like Growth Factor-Binding Protein 7 (IGFBP7) and Tissue Inhibitor of Metalloproteinases-2 (TIMP-2)	Cell cycle arrest biomarkers during the early stage of oxidative cell stress	Kashani et al. [73]
Increased thioredoxin in plasma (Trx)	Compensatory response to increased inflammation and oxidative stress	Wang et al. [74]

**Table 2 antioxidants-12-01772-t002:** Drugs associated with oxidative stress and drug-induced acute kidney injury (AKI).

Drug	Pharmacological Class
Cysplatin	Chemoterapeutic agent
Ifosfamide	Chemoterapeutic agent
Pemetrexed	Chemoterapeutic agent
Gentamycin	Antibiotic
Colistin	Antibiotic
Amphotericin B	Antifungal agent
Foscarnet	Antiviral agent
Cidofovir	Antiviral agent
Cyclosporine A	Immunosupressant
Tacrolimus	Immunosupressant
Pamidronate	Biphosphonate
Zoledronic Acid	Biphosphonate
Penicillins	Antibiotic
Cephalosporins	Antibiotic
Quinolones	Antibiotic
Vancomycin	Antibiotic
Omeprazole	Proton Pump Inhibitor
Non-steroidal anti-inflammatory agents	Analgesic, anti-inflammatory, and anti-pyretic

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
