# Peer review of "The Role of Oxidative Stress in Kidney Injury"

_antioxidants, 2023, doi:10.3390/antiox12091772_

Round 1
Reviewer 1 Report
Remarks to the Authors
The topic of the review article antioxidants-2553018 entitled “The role of oxidative stress in kidney injury” is interesting taking into account the growing prevalence of kidney diseases. Recognition of the involvement of oxidative stress in the development of kidney diseases may shed new light on the protection and treatment of kidney diseases. However, numerous questions have to be addressed.
The main questions that should be addressed are mentioned below.
Point-by-point remarks to the Authors
1) The Abstract section needs rewriting. The abstract should clearly define the purpose of the work, indicate its novelty, provide the most important conclusions and specify the outlook.
2) There is no introduction justifying the desirability of the Authors to undertake the review of the literature on the subject of the work. It has not been specified what the work can bring new, what it will explain, etc.
3) There is no information on the literature search strategy (PRISMA). What databases were searched and what keywords were used?
4) Some unscientific phrases are used in the manuscript. For example “increased oxidative stress” is used in the whole manuscript. Oxidative stress is a state of imbalance between the processes of oxidation and reduction with the predominance of the former. It is not a measurable parameter, and thus it can not be “increased”. This state may occur or not and the extent of its severity may be estimated based on the measurement of numerous markers. The same refers to “High levels of oxidative stress”, “reduce oxidative stress”, “less oxidative stress”, “significant oxidative stress”, “a decrease in antioxidative mechanisms”, and “higher values of ROS”.
5) The sentence “Increased oxidative stress is one of the essential factors in kidney injury and catalyzes cardiovascular risk” (lines 12-13) needs correction. Oxidative stress catalyzes cardiovascular risk – too much abbreviation was used.
6) Line 34: “will therefore be” whether “is”. This same remark refers to line 49 (“we will present”).
7) Lines 56-57: “The cells are under constant oxidative stress,…” Oxidative stress is a pathological state in the cells. The processes of oxidation and reduction constantly occur in the cells, but due to antioxidative defense mechanisms under proper conditions there is a balance between these processes, and destroying this balance with the prevalence of the processes of oxidation over reduction results in the development of oxidative stress.
8) Inappropriate abbreviations/symbols have been used for superoxide radical, hydrogen peroxide, oxygen, and hydroxyl radical.
9) Line 114: What did the authors mean when they wrote “In patients with HD, several biomarkers of oxidative stress have been found.”? I think that the Authors mean increased concentrations of markers of oxidative stress.
10) Lines 130-132: How long after the HD session the concentration of ROS is enhanced”. Do the concentrations of ROS normalize between hemodialysis sessions?
11) Lines 145-147: “Measuring oxidative stress markers, such as malondialdehyde, is promising in predicting allograft survival and delayed graft function”. What other markers and in which material may be measured?
12) Line 161: “NO synthetase” whether „NO synthase”?
13) Line 170: “…mitochondrial structure, function and redox in rat kidneys…” redox?
14) Line 184: “oxidase enzymes”?
15) Line 210: “superoxide anion” – an abbreviation has been introduced (although wrong) and should be consequently used throughout the manuscript. The same refers to all abbreviations used in the paper (for example line 260 “malondialdehyde (MDA)” and line 271 “malondialdehyde”)
16) The title of subsection 4. suggests to a reader that biomarkers of oxidative stress in AKI are described and discussed. It should be clearly stated which parameters (and where they should be measured) are useful to detect destroying the oxidative/reductive balance in AKI patients.
17) Line 230: Is serum creatinine a marker of oxidative stress?
18) Line 240: “increased oxidative cell stress”?
19) It seems necessary to modify the title of subsection 5 “Antioxidant properties of SGLT-2 inhibitors and non-steroid mineralocorticoid receptor antagonists” to one that will better reflect its content.
20) The manuscript to too general. Examples confirming the development of oxidative stress in patients with kidney diseases and evidence of the relationship between oxidative stress and kidney diseases should be presented. In addition, attention should be paid to the relationship between the severity of oxidative stress and the degree of kidney damage. The title of the manuscript suggests that the role of oxidative stress in the development of various kinds of kidney injury will be presented and discussed, including the organ damage induced by exposure to various xenobiotics such as medicines and environmental or occupational pollutants. Oxidative stress is one of the main mechanisms of nephrotoxic action of various xenobiotic.
21) Line 269: “dexpression”?
22) Typographical errors are present.
23) Conclusions are very general. This section should contain the Authors’ conclusions from the performed overview of the available data and the outlook.
24) List of references should be prepared strictly according to the Instruction for Authors.
Author Response
- The Abstract section needs rewriting. The abstract should clearly define the purpose of the work, indicate its novelty, provide the most important conclusions and specify the outlook.
Thank you for your comment. We have rewritten the abstract.
- There is no introduction justifying the desirability of the Authors to undertake the review of the literature on the subject of the work. It has not been specified what the work can bring new, what it will explain, etc.
Thank you for this comment. We have added an introduction to better explain the reason why we decided to write this manuscript.
- There is no information on the literature search strategy(PRISMA). What databases were searched and what keywords were used?
Thank you for this comment. We have added literature search strategy into the manuscript, including the used databases and also keywords.
- Some unscientific phrases are used in the manuscript. For example “increased oxidative stress” is used in the whole manuscript. Oxidative stress is a state of imbalance between the processes of oxidation and reduction with the predominance of the former. It is not a measurable parameter, and thus it can not be “increased”. This state may occur or not and the extent of its severity may be estimated based on the measurement of numerous markers. The same refers to “High levels of oxidative stress”, “reduce oxidative stress”, “less oxidative stress”, “significant oxidative stress”, “a decrease in antioxidative mechanisms”, and “higher values of ROS”.
Thank you for this comment. We have corrected this in the manuscript.
- The sentence “Increased oxidative stress is one of the essential factors in kidney injury and catalyzes cardiovascular risk” (lines 12-13) needs correction. Oxidative stress catalyzes cardiovascular risk – too much abbreviation was used.
Thank you for this comment, we have corrected this.
- Line 34: “will therefore be” whether “is”. This same remark refers to line 49 (“we will present”).
Thank you for this comment. This has now been corrected.
- Lines 56-57: “The cells are under constant oxidative stress,…” Oxidative stress is a pathological state in the cells. The processes of oxidation and reduction constantly occur in the cells, but due to antioxidative defense mechanisms under proper conditions there is a balance between these processes, and destroying this balance with the prevalence of the processes of oxidation over reduction results in the development of oxidative stress.
Thank you for this comment, this has now been corrected and made more clear.
- Inappropriate abbreviations/symbols have been used for superoxide radical, hydrogen
peroxide, oxygen, and hydroxyl radical.
Thank you for this comment. We apologize for the errors and have corrected them.
- Line 114: What did the authors mean when they wrote “In patients with HD, several biomarkers of oxidative stress have been found.”? I think that the Authors mean increased concentrations of markers of oxidative stress.
Thank you. Your comment is spot on, that is exactly what we wanted to state, and have now corrected it.
- Lines 130-132: How long after the HD session the concentration of ROS is enhanced”. Do the concentrations of ROS normalize between hemodialysis sessions?
Thank you for this interesting comment/observation. This was difficult information to find, but it appears that the level of ROS falls to its predialysis level by the end of the HD session. This has been added.
- Lines 145-147: “Measuring oxidative stress markers, such as malondialdehyde, is promising in predicting allograft survival and delayed graft function”. What other markers and in which material may be measured?
Thank you. We have added several other markers of oxidative stress in this population.
- Line 161: “NO synthetase” whether „NO synthase”?
Thank you. This has been addressed and corrected into NO synthase.
- Line 170: “…mitochondrial structure, function and redox in rat kidneys…” redox?
Thank you for your comment. This has been corrected.
- Line 184: “oxidase enzymes”?
Thank you for your comment, this has been corrected.
- Line 210: “superoxide anion” – an abbreviation has been introduced (although wrong) and should be consequently used throughout the manuscript. The same refers to all abbreviations used in the paper (for example line 260 “malondialdehyde (MDA)” and line 271 “malondialdehyde”)
Thank you, this has been corrected.
- The title of subsection 4. suggests to a reader that biomarkers of oxidative stress in AKI are described and discussed. It should be clearly stated which parameters (and where they should be measured) are useful to detect destroying the oxidative/reductive balance in AKI patients.
Thank you for your valuable comment. We have rewritten this part of the manuscript and also added another Figure to make things more clear. Additionally, new references were added.
- Line 230: Is serum creatinine a marker of oxidative stress?
No, serum creatinine is a marker of kidney injury. This has now been made clearer in the manuscript.
- Line 240: “increased oxidative cell stress”?
This has now been corrected.
- It seems necessary to modify the title of subsection 5 “Antioxidant properties of SGLT-2 inhibitors and non-steroid mineralocorticoid receptor antagonists” to one that will better reflect its content.
Thank you for your valuable comment. We have decided to change the title of this subsection and make it a bit more general.
- The manuscript to too general. Examples confirming the development of oxidative stress in patients with kidney diseases and evidence of the relationship between oxidative stress and kidney diseases should be presented. In addition, attention should be paid to the relationship between the severity of oxidative stress and the degree of kidney damage. The title of the manuscript suggests that the role of oxidative stress in the development of various kinds of kidney injury will be presented and discussed, including the organ damage induced by exposure to various xenobiotics such as medicines and environmental or occupational pollutants. Oxidative stress is one of the main mechanisms of nephrotoxic action of various xenobiotic.
We have added additional information and data on oxidative stress in rhabdomiolysis-induced AKI, diabetic kidney disease, ADPKD and also FSGS. We believe that by adding this data we have enhanced the manuscript and also made it less general and more focused on the role of oxidative stress in different forms of kidney injury, including genetic causes.
We have also added a chapter on the role of oxidative stress in xenobiotic nephrotoxicity. Additionally, we also added another Figure (presenting the drugs that are most commonly associated with oxidative stress and drug-induced AKI).
- Line 269: “dexpression”?
This has been corrected.
- Typographical errors are present.
We have corrected these errors to the best of our ability.
- Conclusions are very general. This section should contain the Authors’ conclusions from the performed overview of the available data and the outlook.
Thank you for your comment, we have rewritten the conclusion.
- List of references should be prepared strictly according to the Instruction for Authors.
Thank you for your comment. We have carefully reviewed and corrected references
Reviewer 2 Report
I read with interest the manuscript submitted to me for review, which fully narrates the mechanisms of acute renal failure induced by oxidative stress.
Apart from this, among the causes of AKI and in particular of acute tubular necrosis from intrarenal factors, both occupational and environmental toxic substances are completely forgotten (in particular metals such as Hg, Cr, Cd, Pb or solvents such as tri and tetrachloroethylene, hexachloro-1 :3-butadiene). In the literature there are copious examples, above all experimental, but also on human. I suggest adding a short chapter that takes into account the importance of these substances.
Author Response
I read with interest the manuscript submitted to me for review, which fully narrates the mechanisms of acute renal failure induced by oxidative stress. Apart from this, among the causes of AKI and in particular of acute tubular necrosis from intrarenal factors, both occupational and environmental toxic substances are completely forgotten (in particular metals such as Hg, Cr, Cd, Pb or solvents such as tri and tetrachloroethylene, hexachloro-1 :3-butadiene). In the literature there are copious examples, above all experimental, but also on human. I suggest adding a short chapter that takes into account the importance of these substances.
Response: Thank you very much. We have added a chapter on the role of occupational and environmental toxic substances in oxidative stress and kidney injury.
Reviewer 3 Report
Review of the manuscript entitled: The role of oxidative stress in kidney injury. The manuscript is interesting and summarizes the current state of knowledge in topic of the role of oxidative stress in kidney injury. but corrections need to be made.
In abstract and introduction (1. Acute kidney injury) clear aim of the manuscript should be added e.g. "The aim of the present study was to ..." e.g. 49-50 lines. Moreover introduction must be ended with the aim of the manuscript.
References should be added to 20, 21, 38, 144, 170, 195, 224, lines.
In my opinion the manuscript is very well prepared. However, I would suggest adding a table summarizing the studies from the references. The table will facilitate the analysis of research.
Author Response
Review of the manuscript entitled: The role of oxidative stress in kidney injury. The manuscript is interesting and summarizes the current state of knowledge in topic of the role of oxidative stress in kidney injury. but corrections need to be made.
In abstract and introduction (1. Acute kidney injury) clear aim of the manuscript should be added e.g. "The aim of the present study was to ..." e.g. 49-50 lines. Moreover, introduction must be ended with the aim of the manuscript.
References should be added to 20, 21, 38, 144, 170, 195, 224, lines.
Response: Thank you for your valuable comment. We have added an introduction, made the aim of the manuscript clearer and also added several new references.
In my opinion the manuscript is very well prepared. However, I would suggest adding a table summarizing the studies from the references. The table will facilitate the analysis of research.
Response: Thank you for your comment. We have added two new figures summarizing the data.
Round 2
Reviewer 1 Report
The revised review paper antioxidants-2553018 entitled “The role of oxidative stress in kidney injury” has been markedly improved. The Authors responded to all remarks; however, some questions need attention and should be addressed. The questions that should be addressed are mentioned below.
The questions that should be addressed:
- Lines 18-19; The sentence “Additionally, the role of oxidative stress in the development of xenobiotic nephrotoxicity and also in the development after exposure to various environmental and occupational pollutants is presented.”
- The Authors added two Tables; however, named them Figures (Figure 3 and Figure 4). It is necessary to change Figures 3 and 4 into Table 1 and Table 2.
- In the table presenting the main markers of oxidative stress in acute kidney injury, the first column (Authors) should be the last one. Moreover, in the column “The role of biomarkers”, a different font was used.
- Line 371; I think, the Authors mean toxic substances, not toxins. Toxin is a toxic substance of natural origin only.
- The added section 9 “Occupational and environmental toxic substances, oxidative stress, and kidney injury” is too general.
Author Response
Dear Reviewer,
thank you very much for your hard work on improving our manuscript. We have addressed all your comments, and our responses are stated below.
- Lines 18-19; The sentence “Additionally, the role of oxidative stress in the development of xenobiotic nephrotoxicity and also in the development after exposure to various environmental and occupational pollutants is presented.”
Thank you, although we are unsure what should be changed in this sentence. We have changed xenobiotic to drug-related, making this terminology clearer. Hopefully, this is now OK.
- The Authors added two Tables; however, named them Figures (Figure 3 and Figure 4). It is necessary to change Figures 3 and 4 into Table 1 and Table 2.
Thank you very much. This has now been corrected.
- In the table presenting the main markers of oxidative stress in acute kidney injury, the first column (Authors) should be the last one. Moreover, in the column “The role of biomarkers”, a different font was used.
Thank you. This has now been corrected.
- Line 371; I think, the Authors mean toxic substances, not toxins. Toxin is a toxic substance of natural origin only.
Thank you, this has also been corrected into toxic substances.
- The added section 9 “Occupational and environmental toxic substances, oxidative stress, and kidney injury” is too general.
Thank you. We have added some new data on molecular mechanisms of nephrotoxicity after exposure to several heavy metals. Additionally, we added new references.
Reviewer 2 Report
The authors have fully reply to my suggestion.
I have not further comments.
Author Response
Thank you very much.